# Novel Variant *IMPDH1* c.134A>G, p.(Tyr45Cys): Phenotype–Genotype Correlation Revealed Likely Benign Clinical Significance

**DOI:** 10.3390/ijms241511889

**Published:** 2023-07-25

**Authors:** Mirjana Bjeloš, Ana Ćurić, Mladen Bušić, Benedict Rak, Biljana Kuzmanović Elabjer, Leon Marković

**Affiliations:** 1University Eye Department, Reference Center of the Ministry of Health of the Republic of Croatia for Inherited Retinal Dystrophies, Reference Center of the Ministry of Health of the Republic of Croatia for Pediatric Ophthalmology and Strabismus, University Hospital “Sveti Duh”, 10000 Zagreb, Croatia; dr.mbjelos@gmail.com (M.B.); akrizanovic25@gmail.com (A.Ć.); benedict.rak@gmail.com (B.R.); belabjer@kbsd.hr (B.K.E.); lemarkovic@gmail.com (L.M.); 2Faculty of Medicine, Josip Juraj Strossmayer University of Osijek, 31000 Osijek, Croatia; 3Faculty of Dental Medicine and Health Osijek, Josip Juraj Strossmayer University of Osijek, 31000 Osijek, Croatia

**Keywords:** retina, ocular disorder, novel mutations, genetic mutation

## Abstract

Pathogenic variants in *IMPDH1* are associated with autosomal dominant retinitis pigmentosa 10 (RP10), and Leber congenital amaurosis 11. This case report of a 13-year-old girl with Down’s syndrome and keratoglobus is aimed at linking the novel variant *IMPDH1* c.134A>G, p.(Tyr45Cys), a variant of uncertain significance, to a clinical phenotype and to provide grounds for the objective assignment of its benign features. RP10 is characterized by the early onset and rapid progression of ocular symptoms, beginning with nyctalopia in childhood, accompanied by typical RP fundus changes. As evidenced via thorough clinical examination and testing, none of the RP10 characteristics were present in our patient. On the contrary, our patient who was heterozygous for *IMPDH1* c.134A>G, p.(Tyr45Cys) showed no signs of peripheral retinal dystrophy, and did not manifest any disease characteristics typical of the *IMPDH1* gene mutation. Consequently, we conclude that the variant did not contribute to the phenotype. According to standards and guidelines for the interpretation of sequence variants, *IMPDH1* c.134A>G, p.(Tyr45Cys) revealed likely benign features.

## 1. Introduction

The *IMPDH1* gene encodes an inosine-5-prime-monophosphate dehydrogenase that acts as a homotetramer in order to regulate cell growth. This enzyme catalyzes the rate-limiting step in the de novo synthesis of guanine nucleotides, and within photoreceptors it is thought to play an important role in cyclic nucleotide metabolism [1]. Pathogenic variants in *IMPDH1* are associated with autosomal dominant retinitis pigmentosa 10 (RP10), and Leber congenital amaurosis 11 (LCA11) [2,3].

An early onset and rapid progression of ocular symptoms, starting in childhood with nyctalopia, followed by visual field constriction are typical features of RP10 [2].

The biggest barrier to interpreting genetic test results is the presence of variants of uncertain significance (VUSs), i.e., genetic variants that are of uncertain pathogenicity [4]. These VUSs populate most genetic testing reports and increase with testing panel growth and with the implementation whole genome sequencing [5]. This is the first report of the novel variant *IMPDH1* c.134A>G, p.(Tyr45Cys), a VUS that is autosomal-dominant. This case report is aimed at linking the variant *IMPDH1* c.134A>G, p.(Tyr45Cys) to a clinical phenotype and to provide grounds for the objective assignment of its benign features in order to enable a clinical interpretation of genetic test results.

## 2. Case Presentation

### 2.1. Case Description

A 13-year-old girl with Down’s syndrome (DS) was referred to our Reference Center for Inherited Retinal Dystrophies for clinical examination and genetic testing. At birth, she was diagnosed with DS, esophageal atresia, and congenital hypothireosis. At the age of 6 months, she underwent heart surgery due to a congenital heart defect. Due to hearing loss, aids were provided. The patient was diagnosed with high myopia and amblyopia at the age of 4. Seemingly ‘simple’ high myopia and hearing loss prompted a consideration of possible Usher’s syndrome.

She was enrolled in an ordinary elementary school and an individual education plan with a teaching assistant was provided. In June and October 2022, a crosslinking of both eyes was performed due to keratoglobus.

On clinical examination, her best-corrected visual acuity (BCVA) tested with the ETDRS (Early Treatment Diabetic Retinopathy Study) chart was 0.8 logMAR on the right eye and 0.7 logMAR on the left eye, tested at 4 m. Tested at 40 cm, the BCVA measured 0.4 logMAR and 0.5 logMAR on the right eye and left eye. She manifested right-eye intermittent exotropia with local stereopsis of 100 s of arc. CSV- 1000 contrast sensitivity for the spatial frequencies of 3, 6, 12 and 18 cpd., tested binocularly, was reduced to 1.49 logMAR, 1.55 logMAR, 1.08 logMAR, and 0.64 logMAR, respectively.

Monocular contrast vision was also reduced measuring 1.17 logMAR, 1.55 logMAR, 1.25 logMAR and 0.64 logMAR on both eyes. Farnsworth’s D-15 dichotomous test was unremarkable, while the Lanthony desaturated 15-hue panel showed discrete pathological color discrimination, probably of the diffuse type. Octopus^®^ (Haag-Streit Inc, Mason, OH, USA) G and M standard TOP static perimetry revealed reduced retinal sensitivity in central 30° and 12°: a mean sensitivity of 14.6 dB (30°) and 24.2 dB (12°) on the right eye and those of 13.3 dB (30°) and 19.4 (12°) on the left eye were achieved. Goldmann kinetic perimetry tested with the III4e stimulus revealed a visual field of normal width with a total sum of meridians of 1449° and 1452° on the right eye and the left eye, respectively. MAIA microperimetry (iCare Finland Oy, Vantaa, Finland) revealed abnormal macular integrity with an average threshold of 14.1 dB on the RE and that of 20.2 dB on the left eye. Fixation was unstable on the right eye (P1 = 63%; P2 = 75%), and stable on the left eye (P1 = 81%; P2 = 91%).

HRA+OCT (optical coherence tomography) Spectralis^®^ imaging (Heidelberg Engineering, Heidelberg, Germany) depicted a myopic configuration of central 30° but normal stratification of the retina (Figure 1).

Optos^®^ California (Optos Inc., Marlborough, MA, USA) ultra-widefield imaging depicted a eutrophic optic nerve head (ONH) on both eyes, with peripapillary ring atrophy, which was more pronounced on the left eye. Posterior staphyloma was evident, but the macular architecture was unremarkable, with homogeneous pigmentation (Figure 2). Vascularization reached the periphery. Fundus autofluorescence (FAF) revealed a normal pattern of attenuated signal centrally with a relatively homogeneous distribution of retinal pigment epithelium (RPE) lipofuscin outside the macula (Figure 3).

Pattern reversal visual evoked potential (p-VEP) testing (Roland Consult RETI-port/scan 21, Roland Consult Stasche and Finger GmbH–German Engineering, Brandenburg an der Havel, Germany) performed in line with ISCEV (International Society for Clinical Electrophysiology of Vision) standards using goldcup electrodes revealed waveforms of reduced amplitudes elicited with a 0.25° checker on both eyes (Figure 4).

Full-field electroretinography testing (FFERG) (Roland Consult RETI-port/scan 21, Roland Consult Stasche and Finger GmbH–German Engineering, Brandenburg an der Havel, Germany) according to ISCEV standards using 3M electrodes (3M, Saint Paul, MN, USA) exhibited scotopic responses on both eyes within the range of normal (Figure 5A–H). During full-field photopic stimulus recording, the patient experienced significant discomfort due to photophobia that affected the amplitudes of waveforms (Figure 5I–L). 

IOLMaster^®^ 700 (Zeiss, Oberkochen, Germany) optical biometry measured axial length values of 25.72 mm of the right eye and 26.04 mm of the left eye, and keratometry values of K1 = 50.55 D ax 3° and K2 = 51.86 D ax 93° for the right eye, and K1 = 50.25 D ax 27° and K2 = 51.32 ax 117° for the left eye. Thus, high myopia of −11.50 Dsph on the right eye, and −12 Dsph/−0.75 Dcyl ax 70° on the left eye as demonstrated via retinoscopy could be described as predominantly refractive. 

Full-field stimulus testing (Metrovision, Perenchies, France) detected a response of 55 dB, non-recordable dB, and 64 dB to white, red and blue light, respectively on right eye, and of 74 dB, 55 dB, and 74 dB on the left eye. Pupillometry was within the normal range.

Given the clinical suspicion of inherited retinal dystrophy (IRD), the patient was referred to genetic testing. Prior to testing, written informed consent was obtained from the patient’s legal guardian. A saliva sample was collected and sequence analysis using the Blueprint Genetics Retinal Dystrophy Panel Plus (version 7, 30 October 2021) identified a heterozygous missense variant, *IMPDH1* c.134A>G, p.(Tyr45Cys), labeled as VUS. 

### 2.2. Genetic Testing Methodology

Blueprint Genetics’ Plus Analysis is a combination of both sequencing and deletion/duplication (copy number variant (CNV)) analysis of 314 nuclear and 37 mitochondrial genes. The target region for each gene includes coding exons and ±20 base pairs from the exon–intron boundary.

Laboratory process: The total genomic deoxyribonucleic acid (DNA) was extracted from the biological sample using a bead-based method. DNA quality and quantity were assessed using electrophoretic methods at Blueprint Genetics (Blueprint Genetics, Espoo, Finland). After the assessment of DNA quality, the qualified genomic DNA sample was randomly fragmented using non-contact, isothermal sonochemistry processing. A sequencing library was prepared by ligating sequencing adapters to both ends of the DNA fragments. Sequencing libraries were size-selected with the bead-based method to ensure an optimal template size and amplified via a polymerase chain reaction (PCR). Regions of interest (exons and intronic targets) were targeted using the hybridization-based target capture method. The quality of the completed sequencing library was controlled by ensuring a correct template size and quantity to eliminate the presence of leftover primers and adapter–adapter dimers. Ready sequencing libraries that passed quality control were sequenced using Illumina’s sequencing-by-synthesis method using paired-end sequencing (150 by 150 bases). Primary data analysis converting images into base calls and associated quality scores was carried out with the sequencing instrument using Illumina’s proprietary software (version 3.4.10.732). All steps were performed at Blueprint Genetics.

Quality control: The sequencing run included in-process reference sample(s) for quality control, which passed the Blueprint Genetics’ thresholds for sensitivity and specificity. The patient’s sample was subjected to thorough quality control measures including assessments for contamination and sample mix-ups. CNVs, defined as single-exon or larger deletions or duplications, were detected from the sequence analysis data using a proprietary bioinformatic pipeline. The difference between observed and expected sequencing depth in the targeted genomic regions was calculated, and regions were divided into segments with variable DNA copy numbers. The expected sequencing depth was obtained using other samples processed in the same sequence analysis as a guiding reference. The sequence data were adjusted to account for the effects of varying guanine and cytosine content. Bioinformatics and quality control processes were performed by Blueprint Genetics.

Databases: The pathogenic potential of the identified variants was assessed by considering the predicted consequence of the change, the degree of evolutionary conservation and the number of reference population databases and mutation databases such as, but not limited to, the gnomAD, ClinVar, HGMD (Human Gene Mutation Database) Professional and Alamut Visual. In addition, the clinical relevance of any identified CNV was evaluated by reviewing the relevant literature and databases.

Confirmation of sequence alterations: Sequence variants classified as pathogenic, likely pathogenic and VUS were confirmed using bi-directional Sanger sequencing when they did not meet Blueprint Genetics’ stringent next-generation sequencing (NGS) quality metrics for a true positive call. The confirmation of sequence alterations was performed at Blueprint Genetics.

Confirmation of copy number variants: CNVs were confirmed using a digital PCR assay if they covered less than 10 exons (heterozygous), less than 3 exons (homo/hemizygous) or were not confirmed at least three times previously at the Blueprint Genetics laboratory. Furthermore, CNVs of any size were not confirmed when the breakpoints of the call could be determined. The confirmation of copy number variants was performed at Blueprint Genetics.

Analytic validation: The detection performance of this panel was expected to be in the same range as that of the high-quality, clinical grade NGS sequencing assay used to generate the panel data (nuclear DNA: sensitivity for single-nucleotide variants, 99.89%; indels 1–50 bps, 99.2%, one-exon deletion, 100%; five-exons CNV, 98.7%; and specificity, >99.9% for most variant types). It does not detect very low-level mosaicism as a a variant with a minor allele fraction of 14.6% can be detected in 90% of cases.

## 3. Discussion

### 3.1. IMPDH1 Gene

*IMPDH* genes are highly conserved genes found in virtually all species [6]. *IMPDH1*, a ubiquitously expressed enzyme in most human tissues is a homotetramer, that acts as a catalyzer in the rate-limiting step in the de novo synthesis of guanine nucleotides [1]. Monomers are composed of an eight-stranded α/β barrel with catalytic activity and two CBS regions constituting a flanking subdomain [6]. Mutations in *IMPDH1* have only been detected in retinopathies, indicating its important role in cyclic nucleotide metabolism within photoreceptors [1,6]. Moreover, *IMPDH1* levels are higher in both rods and cones compared to those in most other tissues underlining a unique requirement of *IMPDH1* in photoreceptors [6]. In human eyes, a canonical *IMPDH1* form has not been detected [7]. New data reported on the increased catalytic activity of *IMPDH1* retinal isoforms compared to that of the canonical variant [8]. 

The mutation frequency of the *IMPDH1* gene was approximately 2–5% of that of the autosomal-dominant RP cases [6,9,10,11]. De novo *IMPDH1* variants were also identified rarely in isolated LCA [11]. The disease-associated variants do not affect the enzyme activity of IMPDH1, but alter the single-stranded nucleic acid binding property. Bowne et al. identified an *IMPDH1* missense variant, c.676G>A, p.(Asp226Asn), in three autosomal-dominant RP families [9]. Additionally, another missense variant, *IMPDH1* c.802G>A, p.(Val268Ile), was observed in 1 of a cohort of 60 autosomal-dominant RP families. Further variants in *IMPDH1* have been described in patients with autosomal-dominant RP [11,12,13]. Two *IMPDH1* variants, c.313C>T,p.(Arg105Trp) and c.594 T>G, p.(Asn198Lys), were found in patients with LCA [11]. In the Human Gene Mutation Database (Professional 2022.1), 34 disease-causing variants in *IMPDH1* are reported of which the majority are missense variants [14]. One variant affecting the consensus splice donor site, *IMPDH1* c.402+1G>T, has been reported as homozygous in a RP patient [15].

### 3.2. IMPDH1 c.134A>G, p.(Tyr45Cys)

To the best of our knowledge, this variant has not been reported in the medical literature or in disease-related variation databases. In gnomAD v2.1.1 there is one individual that is heterozygous for this variant and one additional individual that is heterozygous in gnomAD v3.1.2 [16]. The variant is present in one individual as heterozygous in the Bravo whole genome reference database (>130,000 genomes) as well [17]. The variant is predicted to be tolerated by most in silico tools utilized (Polyphen, SIFT). In addition, the alternate amino acid is present at this position in two mammals [18], suggesting that this amino acid change may be tolerated.

### 3.3. Phenotype–Genotype Correlation

Most IRD genes are still linked to uncertain and variable phenotypes. Recently, a complex set of candidate gene variants were defined that might act as modifier genes determining a more precise genotype–phenotype correlation [19]. Furthermore, the epitranscriptomic study of oxidatively stressed RPE cells proved a possible role of ribonucleic acid (RNA) editing in retinal dystrophy development [20]. RP10 is characterized by the early onset and rapid progression of ocular symptoms, beginning with night blindness in childhood, followed by severe visual field constriction and reductions in BCVA [2,21]. 

A Spanish family with RP10 presented a relatively early onset of symptoms (mean age of onset, 12.9 years): night blindness, severe constriction of visual fields, fundus changes characteristic of RP, optic disc pallor, retinal vascular attenuation, and bone spicule pigmentary deposits in particular [21].

Coussa et al. reported a 40-year-old French-Canadian man with classic bone spicules, severe fundus and optic nerve pallor, attenuation and straightening of the blood vessels, and bull’s-eye maculopathy with extensive concentric areas of clumped hyperpigmentation. FAF imaging showed central hypopigmented confluent islands surrounded by a hyperpigmented crescent. OCT showed severe foveal dipping, outer retinal tissue thinning, and cystic changes. FFERG demonstrated an equally significant reduction in rod and cone responses [13].

As evidenced by thorough clinical examination and testing, none of these characteristics were present in our patient. On the contrary, our patient showed no signs of peripheral retinal dystrophy. The visual field tested with III4e was normal. ONH was eutrophic, and the vascularization of the retina reached the far periphery (Figure 2). FAF was unremarkable (Figure 3), and FFERG was within the normal range (Figure 5).

Thus, reduced central visual acuity, reduced contrast sensitivity, mild insufficiency in desaturated color vision, microperimetry, automated perimetry findings of central 12° and 30°, and high refractive error are all attributed to keratoglobus, and are not clinical markers of the *IMPDH1* variant.

Keratoglobus is characterized by the generalized thinning and globular protrusion of the cornea [22], resulting in high myopia and an irregular astigmatism that is difficult to treat with refractive correction, as evidenced by the poor visual acuity in our patient. The age of onset is at birth. Unless they experience acute episodes of hydrops and scarring, patients clinically present with clear corneas [22]. 

There are no reports associating corneal ectasia with RP10 and LCA11; however, increasing evidence supports a 100-fold elevated risk of keratoconus in DS patients, presumably caused by variants within or near the *COL6A1* and *COL6A2* genes on chromosome 21 [23,24]. The triplication of the smallest autosome, 21, known as DS, is associated with a variety of ophthalmic conditions [25]. The myopic retinal degeneration, tigroid fundus, and crowding of the retinal vasculature due to early branching found in our patient are consistent with these reports.

Thus, although termed VUS, the variant *IMPDH1* c.134A>G, p.(Tyr45Cys) is observed in a well-documented 13-year-old individual without the signs and symptoms of retinal dystrophy, advocating strong evidence for a benign interpretation. Further, multiple lines of computational evidence suggest no impact on genes or gene products, conforming with the supporting benign criteria [26]. Therefore, combining strong and supporting criteria, the variant *IMPDH1* c.134A>G, p.(Tyr45Cys) could be currently reclassified as likely benign [26].

However, further research is advocated in order to corroborate the findings here observed, because the variant was detected in an individual with trisomy 21.

## 4. Conclusions

The patient here presented is heterozygous for *IMPDH1* c.134A>G, p.(Tyr45Cys), and does not manifest any disease characteristics typical of the *IMPDH1* gene mutation: night blindness in childhood, severe visual field constriction and fundus changes typical for RP. Therefore, in line with the standards and guidelines for the interpretation of sequence variants, a joint consensus recommendation from the American College of Medical Genetics and Genomics and the Association for Molecular Pathology [26], this case report provides evidence of *IMPDH1* c.134A>G, p.(Tyr45Cys) having likely benign features. Consequently, we conclude that the variant did not contribute to the phenotype, and the mutation *IMPDH1* c.134A>G, p.(Tyr45Cys) did not affect the retina of the 13-year-old patient in this particular case. Therefore, we consider the current clinical significance of the variant to be likely benign.

## Figures and Tables

**Figure 1 ijms-24-11889-f001:**
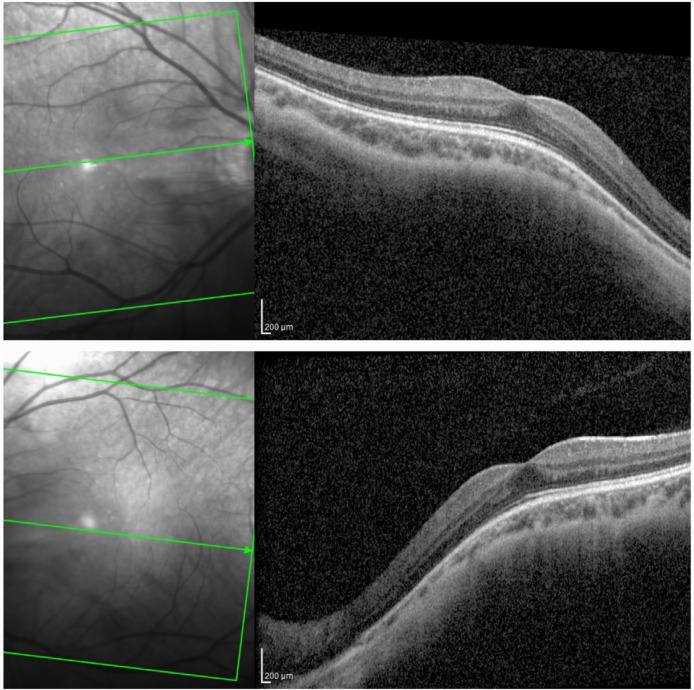
Optical coherence tomography image showing myopic configuration of central 30° and normal stratification of the retina on both eyes (HRA+OCT Spectralis^®^).

**Figure 2 ijms-24-11889-f002:**
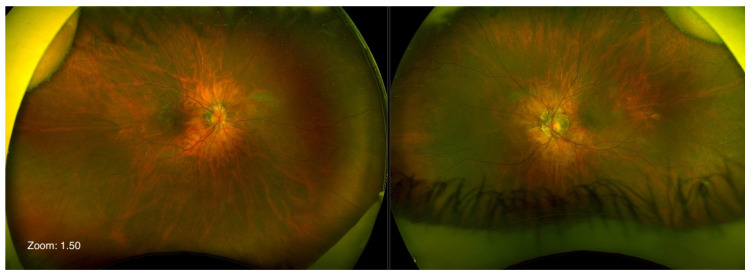
Ultra-widefield image showing eutrophic optic nerve head with peripapillary ring atrophy; more pronounced on the left eye. Posterior staphyloma was evident on both eyes, but macular architecture was unremarkable (Optos^®^ California).

**Figure 3 ijms-24-11889-f003:**
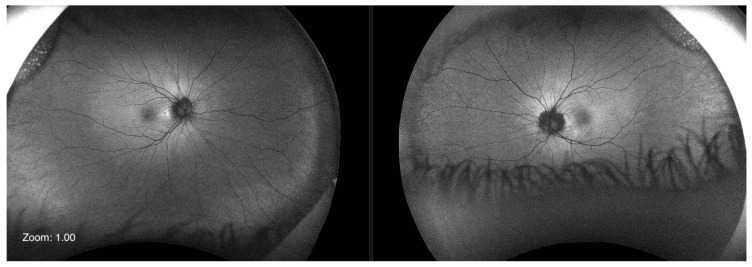
Fundus autofluorescence image depicting normal pattern of attenuated signal centrally with relatively homogeneous distribution of retinal pigment epithelium lipofuscin outside macula on both eyes (Optos^®^ California).

**Figure 4 ijms-24-11889-f004:**
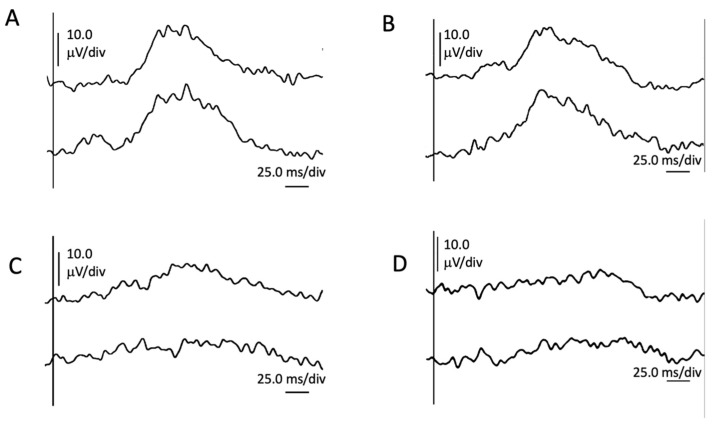
Image showing pattern reversal visual evoked potentials with 1.0° and 0.25° checks of right eye ((**A**,**C**), respectively), and left eye ((**B**,**D**), respectively).

**Figure 5 ijms-24-11889-f005:**
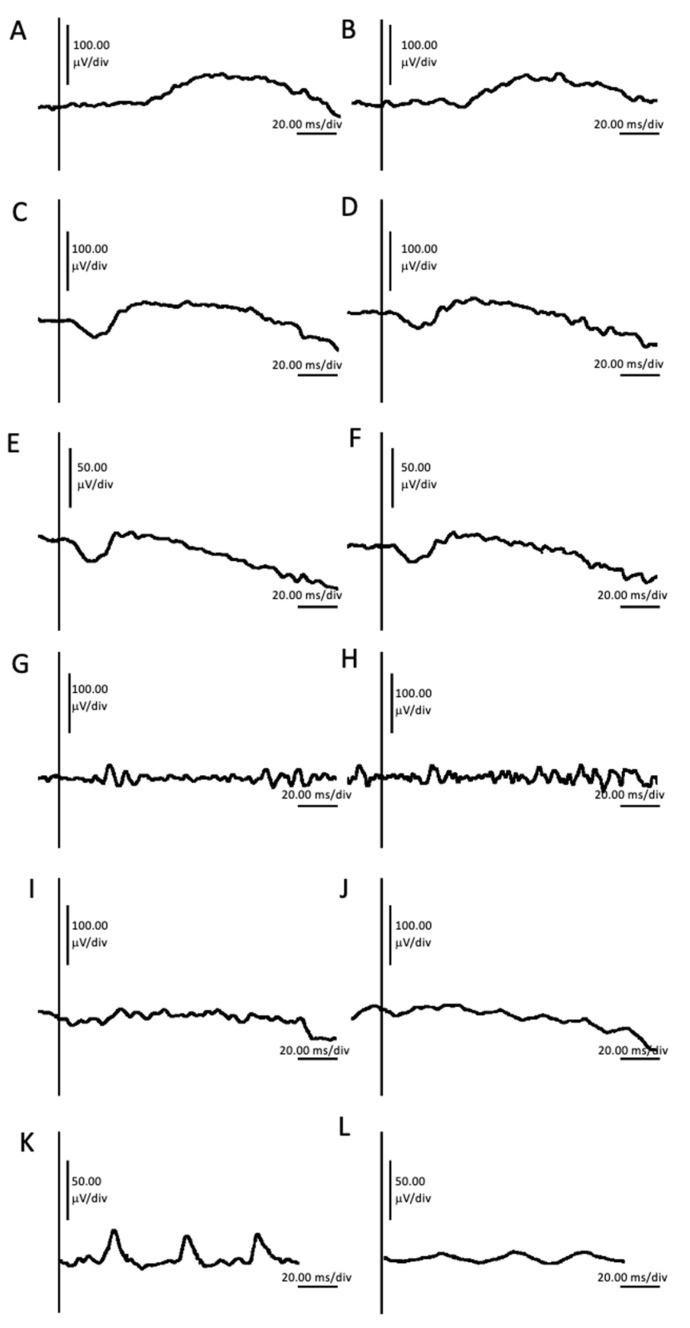
Full-field electroretinography (FFERG) testing of right eye and left eye showing dark-adapted 0.01 ERG of right eye (**A**) and left eye (**B**); dark-dapated 1.0 ERG of right eye (**C**), and left eye (**D**); dark-adapted 3.0 OPs of right eye (**E**) and LE (**F**); dark-adapted 10.0 ERG of right eye (**G**) and left eye (**H**); light-adapted 3.0 ERG of right eye (**I**) and left eye (**J**); and light-adapted 3.0 flicker 30 Hz of right eye (**K**) and left eye (**L**).

## Data Availability

The data presented in this case report are available on request from the corresponding author. The data are not publicity available due to privacy protection.

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
