# Peer review of "Novel Variant IMPDH1 c.134A>G, p.(Tyr45Cys): Phenotype–Genotype Correlation Revealed Likely Benign Clinical Significance"

_ijms, 2023, doi:10.3390/ijms241511889_

Round 1

Reviewer 1 Report

The authors have done a complete ophthalmological study of a 13-year-old patient with Down syndrome who also carries a heterozygous VUS in the IMPDH1 gene, a gene which can cause the RP10 form of retinitis pigmentosa.

The authors conclude that the patient does not have retinal disease. They also conclude that the VUS in IMPDH1 is benign. In my opinion, neither of these two things is publishable, and the second one (status of the VUS) is not proven by medical genetics standards to be benign based on this single, young individual. Unfortunately I cannot think of anything helpful to suggest in terms of editing or improvement.

Reviewer 2 Report

The authors report a case of a young 13-year-old patient with Down syndrome and suspected of presenting hereditary retinal dystrophy. A long ophthalmological description is provided, which does not show the typical clinical signs of retinitis pigmentosa (RP). Conversely, the patient presented with keratoglobus, responsible for the decrease in her central visual acuity, reduced contrast sensitivity, mild insufficiency in desaturated color vision and microperimetry. A IMPDH1 variant c.134A>G has been identified. However, according to the absence of RP signs, this variant should not be classified as a variant of unknown significance, but rather as likely benign.

The manuscript is well written. It is not of major interest but is useful to exclude the involvement of a variant in IMPDH1 gene, which is a difficult gene to interpret because it is rarely identified. Moreover, could the authors add one or more references on the frequency of identification of this gene in the literature (line 133)?

One criticism I could make is that the genetic identification method for this variant is missing. How many genes were tested? By high-throughput sequencing of a panel of genes? by exome? This notion is important because depending on the method, the absence of another pathogenic variant in another gene likely to be responsible for the patient's ophthalmological clinical signs could be excluded.

minor comments:

lane 43: in the age of 6 months is the English correct? should not this be "at the age of "?

lane 44 : comma is missing after "Due to hearing loss"

lane 97 : additional space presents between "right" and "eye"

lane 126 : flanking rather than "flankig"

Reviewer 3 Report

Bjelošet al. realized a very interesting article describing the “Novel Variant IMPDH1 c.134A>G, p.(Tyr45Cys): Phenotype-2 Genotype Correlation Revealed Likely Benign Features”. I consider the manuscript very interesting but, at the same time, I suggest several revisions needed to improve the reliability and the completeness of the paper: 

  1. Background and Introduction: The introduction is generally well-written, but it could benefit from a clearer definition of the problem being investigated. It is recommended to clearly state the research question and its importance in the field of retinal dystrophies.
  2. Novel Variant and Link to Phenotype: The authors should be clearer in explaining how the novel variant IMPDH1 c.134A>G, p.(Tyr45Cys) is linked to the clinical phenotype described. If it's not known, the authors should discuss why they suspect a link.
  3. Case Description: There is a need for clearer and more concise description of the clinical characteristics. For instance, instead of "This child was diagnosed with high myopia and amblyopia at the age of 4.", it could be rewritten as "The patient was diagnosed with high myopia and amblyopia at age 4." Also, jargon and abbreviations should be explained, especially for the first instance.
  4. Methodology: The genetic testing methodology needs more details. It is important to include details such as laboratory procedures, data analysis, quality control measures, and data interpretation. Also, information about ethical approval and patient consent should be added.
  5. Discussion: The discussion should include more comparative analysis with other studies or cases. In this case, how does this novel variant compare with other variants of the IMPDH1 gene associated with retinal dystrophy?
  6. Figures and Legends: All figures need to be referenced in the text. The captions should include more detailed explanations and should stand alone from the main text.
  7. Grammar and Syntax: Several sentences need to be revised for clarity and to correct minor grammar and syntax errors. Also, use of 'RE' and 'LE' for 'right eye' and 'left eye' is not standard in ophthalmology and could be confusing to readers.
  8. Conclusions: The conclusion section is missing. The authors should summarize the main findings, discuss the limitations of the study, and suggest areas for further research.
  9. References: All references should be properly cited in the text. Furthermore, I suggest adding data related to recent bulk transcriptomics studies which could represent a strong substrate to apply the analytic pipeline developed by the authors. The recent PMID: 36290689 and PMID: 36490268 could represent a substrate able to enforce the role of described molecular mechanisms.

The English of the manuscript has to be improved.

Round 2

Reviewer 1 Report

Previous assessment:

"The authors have done a complete ophthalmological study of a 13-year-old patient with Down syndrome who also carries a heterozygous VUS in the IMPDH1 gene, a gene which can cause the RP10 form of retinitis pigmentosa. The authors conclude that the patient does not have retinal disease. They also conclude that the VUS in IMPDH1 is benign. In my opinion, neither of these two things is publishable, and the second one (status of the VUS) is not proven by medical genetics standards to be benign based on this single, young individual. "

While some editing has been done to the manuscript, my previous concerns still exist. The argument that the VUS meets the ACMG standard for "likely benign" is debatable.  By the original published standard it does not meet the criteria - the patient is not a healthy adult and IRDs are notorious for not being completely penetrant. Since 2015 there have been proposed modifications to the standard which, if applied, might get this variant into the likely benign category, mostly based giving more standing to the poor conservation of the amino acid.  

In any case, I believe the appropriate place to present this data is in a submission to ClinVar, not a publication. There is no new knowledge being presented, only a relatively weak variant curation.

Reviewer 3 Report

The authors addressed all suggested points.

The English language requires, now, only minor revisions.